# What Corporate Social Responsibility (CSR) Disclosures Do Chinese Forestry Firms Make on Social Media? Evidence from WeChat

**Ma Zhong [2]**, **Feifei Lu [1],\***, **Yunfu Zhu [2]** and **Jingru Chen [3]**

[1] College of Economics and Management, Qingdao Agricultural University, No. 700 Changcheng Road, Chengyang District, Qingdao 266109, China
[2] College of Economics and Management, Nanjing Forestry University, No. 159 Longpan Road, Xuanwu District, Nanjing 210037, China
[3] School of Economics, University of Bristol, 8 Woodland Road, Bristol BS8 1TN, UK
**\*** Correspondence: feifei.lyu@qau.edu.cn

**Abstract:** Corporate social responsibility (CSR) disclosure serves as a vital bridge for forestry firms to communicate with their stakeholders and obtain legitimacy support. Existing studies focus on forestry firms' CSR disclosures based on CSR reports but lack consideration of such disclosures on social media. In this study, based on WeChat, the most widely used social media platform in China, we obtained 3311 tweets from 36 WeChat Official Accounts (WOA) of 63 Chinese-listed forestry firms in 2018 and used content analysis to classify the CSR information involved in these tweets based on the stakeholder dimensions. The main analysis results show that the top three CSR dimensions disclosed by Chinese forestry firms in social media are the shareholder (28.21%), customer (26.20%), and employee (23.64%) dimensions, and there are also great differences in the subcontent of disclosure concerns in each stakeholder dimension, e.g., approximately 86% of CSR disclosures for customers are product and service information. Additionally, we conducted a content analysis on the CSR reports of forestry firms using WOA. The results show that firms express different concerns in CSR reports than on social media, and the most mentioned dimensions in their reports are the environment (23.69%), employees (20.91%), and shareholders (20.21%). This indicates that there is a significant difference between the stakeholders that Chinese forestry firms focus on in social media and those that they focus on in CSR reports. This paper is the first study to focus on the CSR disclosure of Chinese forestry firms in social media and provides a reference for scholars to understand the information activities of forestry firms in social media.

**Keywords:** ESG; sustainable management; corporate social responsibility (CSR); non-financial disclosure; emerging market

## 1. Introduction

With the launch of the UN 2030 Agenda for Sustainable Development, the United Nations proposed the Global Sustainable Development Goals (SDGs). The 17 SDGs aim to promote better environmental and social performance globally [1]. As the world's largest emerging country, China is actively responding to global development visions and goals, implementing major strategies for carbon peaking and carbon neutrality, and actively assuming international responsibilities. The role of corporate social responsibility (CSR) in business operations and strategic development is becoming increasingly important [2–4]. CSR serves as an important bridge for firms to communicate with stakeholders, help firms to maintain their legitimacy, and obtain key resources. Thus, CSR disclosures are considered by firms to be the most important non-financial information activities [5]. Forests have an irreplaceable integrated value for sustainable economic and ecological development [6–8]; although forested areas only account for approximately 1/3 of the total land

area, the carbon sink storage in forested vegetation areas accounts for almost half of the total terrestrial carbon pool. Forestry enterprises are microeconomic subjects of forestry industry development and are typical resource-sensitive and environment-sensitive enterprises, with the triple attributes of economic, social, and ecological benefits, which are of great significance for sustainable development [9,10].

According to the China Internet Network Information Center (Source: https://m.th epaper.cn/baijiahao_16888548, accessed on 7 June 2022), in 2021, the number of internet users in China was 1.032 billion, of which 1.029 billion were cellular data users, and the internet penetration rate had reached 73.0%. With previously unheard-of access to information and networks, many companies are finding a new voice in their interactions with consumers and other stakeholders. Among them, social media networks such as WeChat and Weibo are developing rapidly, replacing paper media and traditional online media to a large degree. Social media networks are also being used via an increasing number of firms to communicate for CSR [11–14], including forestry firms, who also use them as an important tool for CSR communication. Therefore, this paper conducts a study on CSR information disclosure on social media, which can serve as a new reference for the study of the CSR disclosure of forestry enterprises in various countries.

Our study aims to clarify the CSR disclosure of forestry firms on social media and draw on key differences between traditional media and social media. Related studies only focus on forestry firms' CSR disclosures based on CSR reports but lack consideration of such disclosures on social media. This study is the first to present the viewpoint of the CSR disclosure of Chinese forestry firms in social media and provides a reference to understand the characteristics of the information activities of forestry firms in social media. Potential contributions of this study include: (1) applying a new optic angle to investigate CSR disclosure of forestry firms in China, and extending the point of potential associating and differences between CSR reports and social media; (2) updating and expanding previous study on CSR topic in China, meanwhile, gaining a broader investigation of CSR data from social media; and (3) as the largest emerging market, our study provides a reference understanding of CSR disclosure in other emerging economies for practitioners, managers, and researchers.

The remainder of this paper begins with the literature review and theoretical background about corporate disclosures on social media and CSR disclosure in the forestry industry. Then we turn to the research design, and based on the framework, we describe the variables. The content analysis framework also provides insight into forestry companies activities via a variety of scenarios regarding the level of CSR disclosure. We conclude with a summary of the results with practical implications, limitations, and questions for future research.

## 2. Literature Review and Theoretical Background

### 2.1. Corporate Disclosure on Social Media

Before the advent of social media, corporate disclosure on the internet was of interest to researchers [15–17]. However, corporate online information disclosure in the presocial media era still had some deficiencies, which are similar to those of traditional paper media; for example, information dissemination required review by authorities and third-party platforms and there was a lack of two-way communication with audiences.

With the birth of social media, internet information dissemination has changed dramatically [18]. (1) The "hierarchy" was broken; for example, while traditional media can be divided into "regional", "national", and "international" types, social media does not have such hierarchical properties. (2) The "elite" lost their control, namely, the major "information gatekeepers" (e.g., large newspaper company) lost their control on social media, while non-government organizations (NGOs), consumers, and other weak stakeholders more easily took over the initiative of information spread; (3) social media provides higher dynamism and timeliness, unlike the fixed nature of traditional media; (4) the cost of dissemination of information on social media is extremely low because the use of social media

platforms is free or very inexpensive; (5) information users have a stronger sense of trust of social media because the information dissemination therein is more based on personal relationships and trust rather than commercial speech; (6) social media makes communication much more direct, specifically, it provides more direct dialogue between firms and stakeholders than traditional media; (7) there is a potential for higher rates of public response, namely, information dissemination on social media can obtain more public responses; (8) information on social media achieves wider ranges of diffusion, because social media is not limited to specific information-spreading channels but can spread throughout the social network; (9) information dissemination on social media is subject to less institutional control and involvement but wider public opinion scrutiny.

Social media thus has dramatically changed corporate information activities [19]. First, social media helps to promote the efficiency of corporate information activities and reduce the impact of negative information. For example, Blankespoor, et al. [20] and Prokofieva [14] suggested that social media helps the spread of earnings information and then helps to gain more attention from investors. Yang and Liu [21] found that firms can use social media to mitigate the impact of negative financial information in some ways. In addition, social media also helps stakeholders communicate with each other, which in turn reinforces the monitoring for firms. For example, Ang et al. [22] found that social media plays a complementary governance role in the Chinese market and that stakeholder discussions on social media can effectively curb inefficient corporate mergers and acquisitions.

Social media also plays an important role in CSR information activities [23]. First, social media strengthens CSR communication between firms and their stakeholders [24–26], significantly improves the perception and recognition of CSR among customers, employees, and other stakeholders [27–30], and helps transform CSR into intangible values such as improving corporate reputation [29,31]. It is worth noting that compared to traditional communication, CSR information activities on social media are more indicative, which facilitates impression management for firms. She and Michelon [12] found that firms selectively direct their stakeholders on social media to avoid the proliferation of negative news. Pizzi et al. [32] found that CSR disclosures on social media are selective and not always responsive to stakeholder claims. Russo et al. [33] argued that CSR-oriented firms use social media more as a tool to achieve a higher level of legitimacy than as a tool to manage their sustainability strategies and related performance.

### 2.2. CSR Disclosure in the Forestry Industry

Existing research on CSR disclosure in the forestry industry focuses on traditional reporting vehicles such as CSR reports. First, some studies focus on the information topics in traditional CSR reports for forestry firms. Vidal and Kozak [34] conducted a textual analysis of CSR reports of forestry firms worldwide and found that the proportion of environmental and social dimensions is increasing. Colaço and Simão [35] analyzed CSR disclosures for international forestry firms in the Congo Basin and found that the most frequently disclosed topics are organizational certification and the environment, and that the level of CSR disclosure is higher in firms from Western countries than in those from Asian and African countries. D'Amato et al. [36] analyzed CSR reporting on a global scale involving land use (including forestry firms' CSR reports) and found that its focus was on the sustainability concepts of the circular, green, and bio-economy. Wang and Juslin [37] studied the CSR reports of Chinese forestry enterprises and argued that the most mentioned topic in these reports is economic responsibility. The driving factors of the CSR disclosures of forestry enterprises have also received great interest from scholars. Hansen, et al. [38], Panwar, et al. [39], and Li, et al. [40] studied the drivers of CSR disclosure in cross-country forestry firms based on globalization and firm-level factors. Wang and Juslin [37] analyzed the driving role of managerial characteristics in CSR disclosures. Li, Toppinen, Tuppura, Puumalainen and Hujala [40], and Lu et al. [41] explored the factors influencing CSR disclosure for Chinese forestry firms, focusing on the firm-level factors that characterize forestry firms' CSR disclosure. Lu et al. [42] conducted a study on the impact of managers'

risk awareness on CSR disclosure in Chinese forestry firms. In summary, the research on forestry CSR disclosure only focuses on traditional information vehicles such as CSR reports, and no one discusses their CSR disclosure in emerging social media platforms. Specifically, it is worthwhile studying which kind of topics are preferred by forestry firms in the CSR reporting of social media and the difference between CSR disclosure in social media and traditional reports.

## 3. Research Design

### 3.1. Data Sources

As the largest developing economy, China has a huge forestry industry. According to a report by the China National Forestry and Grassland Administration (Source: http://www.forestry.gov.cn/main/62/20171221/1086586.html, accessed on 10 June 2022), gross output value of forestry reached 6.49 trillion RMB in 2016 and is expected to reach 9 trillion RMB by 2025 (Source: https://news.cgtn.com/news/2022-02-16/China-s-forestry-industry-output-to-reach-9t-yuan-by-2025-17He4nAk6li/index.html, accessed on 10 June 2022). Thus, Chinese forestry firms have a strong representation.

In China, due to regulatory policies, Facebook and Twitter are not available, and similar mainstream social media platforms are WeChat and Weibo. After consideration, we selected the WeChat platform for study. The reasons are as follows:

First, WeChat allows a higher information capacity for users and allows them to provide more substantive information. Compared with Weibo, which requires users to limit their tweets to 140 characters, WeChat does not have such limitations.

Second, WeChat has the largest user base and extremely high user stickiness in China. Its daily logged-in users reached 900 million in 2017, and the monthly active official account (The WeChat platform provides two types of account services: first, a public account, which allows all users to view the tweets of the account at will; the second category, a personal account, is only used by individuals who only want to allow their friends to view the tweets of their account) reached 3.5 million, with nearly 800 million monthly active followers (As a comparison, Weibo had only 376 million monthly active users in 2017 according to the Q3 2017 Sina earnings report. Source: https://www.sohu.com/a/203437993_667510, accessed on 10 June 2022). Moreover, WeChat provides an electronic payment service that is the most widely used by Chinese people (According to WeChat user report in 2018 year, its payment services get 600 million active users. Source: https://www.sohu.com/a/278496021_506058x, accessed on 10 June 2022), and furthermore, creates a high level of dependency and a long-term habit for its users. As a result, the information shared by enterprises through the WeChat Official Account (WOA) will be received by users more reliably and frequently.

The original sample is 63 listed firms in the China forest industry in 2018, and we excluded 27 samples that did not use the WOA or who had used it for less than 1 year. Therefore, there are 36 sample firms included in the analysis. Appendix B presents a list of the sample firms involved in WeChat. Moreover, 11 of 36 forestry firms used in this study also issued CSR reports in that year, which provided us with an opportunity to compare their CSR disclosures in social media channels and traditional channels. We used Python to obtain the WOAs of forestry firms, totaling 3311 articles. CSR reports are sourced from Cninfo (http://www.cninfo.com.cn). Financial and other firm-level data were obtained from CSMAR.

### 3.2. Content Analysis

We used the content analysis system created by Lu, Kozak, Toppinen, D Amato, and Wen [41] (Lu, Kozak, Toppinen, D Amato and Wen [39] design a content analysis system for Chinese forestry enterprises based on the guidance of CASS 3.0 and CNFPIA 2.0, which are released by Chinese Academy of Social Sciences and Chinese National Forestry Products Industry Association), which is for Chinese forestry firms, to categorize the tweets and CSR reports.

The system of Lu, Kozak, Toppinen, D Amato, and Wen [41] contains seven level-1 dimensions based on stakeholder theory: environment, customer, employee, supplier, community, government, and shareholder. Each level-1 dimension contains five to thirteen level-2 dimensions; for example, the employee dimension contains a total of seven level-2 dimensions: "Abidance by rule and laws (EM1)", "Percent of contract signing (EM2)", "Coverage of social insurance (EM3)", "Equal employment institution (EM4)", "Staff development training (EM5)", "Occupational health and safe producing (EM6)", "Staff relation management (EM7)" and "Other employee-related (EM0)". We scored according to the level-2 dimensions of this system; specifically, when the qualitative information of this dimension is provided in the tweet or CSR report, we scored "1", and when the quantitative information is provided, the score was "2". Then, we aggregated the scores of all level-2 dimensions to form level-2 variables. Finally, the level-2 dimension variables were aggregated to level-1 dimensions to form level-1 variables. Based on the above steps, we finally made 16 level-1 variables (eight variables for analysis on tweets and eight variables for analysis on CSR reports, variable definitions are reported in Table 1) and 42 level-2 variables (Since our interest of this paper is not the content of the CSR report, we only use level-2 subdimension variables for CSR disclosure on social media, and no longer use level-2 variables for the CSR report). The scoring index system and examples are reported in Appendix A. The software used for the content analysis is Atlas.ti 8.0.

**Table 1.** Variable definitions.

| Variable | Level Annotation | Definition |
|---|---|---|
| Content variables for CSR disclosure on social media | | |
| WeChat_All | Summary of level-1 variable for social media | Total level of CSR disclosure on social media, equal to the sum of seven level-1 content variable for tweets. |
| ep_w | Level-1 variable for social media | The level of the social media environment dimension CSR disclosure, equal to the sum disclosure scores of all subdimensions of the environment on social media. |
| cu_w | Level-1 variable for social media | The level of the social media customer dimension CSR disclosure, equal to the sum disclosure scores of all subdimensions of the customer on social media. |
| em_w | Level-1 variable for social media | The level of the social media employee dimension CSR disclosure, equal to the sum disclosure scores of all subdimensions of the employee on social media. |
| su_w | Level-1 variable for social media | The level of the social media supplier dimension CSR disclosure, equal to the sum disclosure scores of all subdimensions of the supplier on social media. |
| co_w | Level-1 variable for social media | The level of the social media community dimension CSR disclosure, equal to the sum disclosure scores of all subdimensions of the community on social media. |
| go_w | Level-1 variable for social media | The level of the social media government dimension CSR disclosure, equal to the sum disclosure scores of all subdimensions of the government on social media. |
| sh_w | Level-1 variable for social media | The level of the social media shareholder dimension CSR disclosure, equal to the sum disclosure scores of all subdimensions of the shareholder on social media. |

**Table 1.** *Cont.*

| Variable | Level Annotation | Definition |
|---|---|---|
| ep1_w/ep2_w/ep4 _w/ep5_w/ep6_w/ep7 _w/ep8_w/ep9_w/ep10 _w/ep11_w/ep12_w/ep0_w | Level-2 variable for social media | The level of environment disclosure of the level-2 subdimension on social media, equal to the sum of the scores of the information of the level-2 subdimension of the environment. For example, the variable ep1_w is equal to the sum of the qualitative and quantitative scores of the level-2 subdimension indicator EP1 "Environment management system". The description of the level-2 subdimension indicators are shown in |
| cu1_w/cu2_w/cu4 _w/cu6_w/cu0_w | Level-2 variable for social media | The level of customer disclosure of the level-2 subdimension on social media, equal to the sum of the scores of the information of the level-2 subdimension of the customer. For example, the variable cu1_w is equal to the sum of the qualitative and quantitative scores of the level-2 subdimension indicator CU1 "Product quality management system". The description of level-2 subdimension indicator are shown in |
| em1_w/em3_w/ep4 _w/em5_w/ep6_w/ep7 _w/em0_w | Level-2 variable for social media | The level of employee disclosure of the level-2 subdimension on social media, equal to the sum of the scores of the information of the level-2 subdimension of the employee. For example, the variable em1_w is equal to the sum of the qualitative and quantitative scores of the level-2 subdimension indicator EM1 "Abidance by rule and laws". The description of level-2 subdimension indicators are shown in |
| su1_w/su4_w/su5 _w/su6_w/su0_w | Level-2 variable for social media | The level of supplier disclosure of the level-2 subdimension on social media, equal to the sum of the scores of the information of the level-2 subdimension of the supplier. For example, the variable su1_w is equal to the sum of the qualitative and quantitative scores of the level-2 subdimension indicator SU1 "Responsibility purchasing system". The description of level-2 subdimension indicators are shown in |
| co1_w/co4_w/co6 _w/co0_w | Level-2 variable for social media | The level of community disclosure of the level-2 subdimension on social media, equal to the sum of the scores of the information of the level-2 subdimension of the community. For example, the variable co1_w is equal to the sum of the qualitative and quantitative scores of the level-2 subdimension indicator CO1 "The effect of enterprise operation on community". The description of level-2 subdimension indicators are shown in |
| go1_w/go2_w/go4 _w/go0_w | Level-2 variable for social media | The level of government disclosure of the level-2 subdimension on social media, equal to the sum of the scores of the information of the level-2 subdimension of the government. For example, the variable go1_w is equal to the sum of the qualitative and quantitative scores of the level-2 subdimension indicator GO1 "Enterprise management abided by rule". The description of level-2 subdimension indicators are shown in |
| sh1_w/sh2_w/sh3 _w/sh4_w/sh0_w | Level-2 variable for social media | The level of shareholder disclosure of the level-2 subdimension on social media, equal to the sum of the scores of the information of the level-2 subdimension of the shareholder. For example, the variable sh1_w is equal to the sum of the qualitative and quantitative scores of the level-2 subdimension indicator SH1 "Investor relation management". The description of level-2 subdimension indicators are shown in |

**Table 1.** *Cont.*

| Variable | Level Annotation | Definition |
|---|---|---|
| Content variables for CSR disclosure on CSR report | | |
| Report_All | Summary of level-1 variable for the CSR report | Total level of CSR disclosure on the CSR report, equal to the sum of the seven level-1 content variables for the CSR report. |
| ep_r | Level-1 variable for CSR report | The level of CSR report environment dimension CSR disclosure, equal to the sum disclosure scores of all subdimensions of the environment on social media. |
| cu_r | Level-1 variable for CSR report | The level of the CSR report customer dimension CSR disclosure, equal to the sum disclosure scores of all subdimensions of the customer on social media. |
| em_r | Level-1 variable for CSR report | The level of the CSR report employee dimension CSR disclosure, equal to the sum disclosure scores of all subdimensions of the employee on social media. |
| su_r | Level-1 variable for CSR report | The level of the CSR report supplier dimension CSR disclosure, equal to the sum disclosure scores of all subdimensions of the supplier on social media. |
| co_r | Level-1 variable for CSR report | The level of the CSR report community dimension CSR disclosure, equal to the sum disclosure scores of all subdimensions of the community on social media. |
| go_r | Level-1 variable for CSR report | The level of the CSR report government dimension CSR disclosure, equal to the sum disclosure scores of all subdimensions of the government on social media. |
| sh_r | Level-1 variable for CSR report | The level of the CSR report shareholder dimension CSR disclosure, equal to the sum disclosure scores of all subdimensions of the shareholder on social media. |

### 3.3. Descriptions of Variables

Table 1 reports the variables used in the analysis of this paper:

(1) Content variables for CSR disclosure on social media. These include the total social media disclosure variable WeChat_All, social media environment dimension disclosure variable ep_w, social media customer dimension disclosure variable cu_w, social media employee dimension disclosure variable em_w, social media supplier dimension disclosure variable su_w, social media community dimension disclosure variable co_w, social media government dimension disclosure variable go_w, and social media shareholder dimension disclosure variable sh_w. Moreover, we defined forty-two level-2 subdimension variables, e.g., variable ep1_w stands for the reporting level of the social media environment EP1 subdimension (Some of the level-2 subdimensions indicators of [41], 10 dimensions in all, among them the forest biodiversity conservation dimension EP3 and dispute resolution mechanism dimension CU3, are not addressed in the social media disclosure, so we do not create variables for these dimensions).

(2) Content variables for CSR disclosures in CSR reports. These also include eight variables: the total CSR report disclosure variable Report_All, CSR report environment dimension disclosure variable ep_r, CSR report customer dimension disclosure variable cu_r, CSR report employee dimension disclosure variable em_r, CSR report supplier dimension disclosure variable su_r, CSR report community dimension disclosure variable co_r, CSR report government dimension disclosure variable go_r, and CSR report shareholder dimension disclosure variable sh_r.

## 4. Results

### *4.1. Content Analysis of CSR Disclosures on Social Media*

4.1.1. Total and Level-1 Dimension Analysis on Social Media

Table 2 reports the descriptive statistics of the overall and level-1 dimension variables of social media CSR disclosure for 36 forestry firms. The mean value of the overall disclosure variable WeChat_All is 103.3, and the median value is 72.5. Rows (2) to (8) of Table 2 show the statistics of the environment dimension variable ep_w, customer dimension variable cu_w, employee dimension variable em_w, supplier dimension variable su_w, community dimension variable co_w, government dimension variable go_w, and shareholder dimension variable sh_w, where the top three highest disclosure dimensions are shareholder, customer, and employee, respectively. The mean (median) values of the shareholder dimension variable sh_w, customer dimension variable cu_w, and employee dimension variable em_w are 29.14 (18.5), 27.06 (8.5), and 24.42 (5), respectively. The last four variables are the government dimension variable go_w, environment dimension variable ep_w, community dimension variable co_w, and supplier dimension variable su_w, with mean (median) values of 8.78 (3), 8.14 (2), 5.17 (3), and 0.64 (0), respectively. Column (4) shows the proportion of subdimension disclosure to total disclosure. Among them, the top three dimensions with the highest proportion of disclosure are shareholders, customers and employees, with 28.21%, 26.20%, and 23.64% (The information proportion of level–1 subdimension is equal to the mean value of the subdimension indicator divided by the mean value of the total rating indicator (WeChat_All), for example, the mean value of the environment dimension variable ep_w is 8.14, divided by the mean value of the total rating indicator WeChat_All, 103.3, which equals 7.88%), respectively, which cumulatively account for approximately 80% of the total CSR disclosure on social media. The above results show that the stakeholders that forestry firms care most about in their social media CSR disclosures are shareholders, customers, and employees, and these three stakeholders account for approximately 80% of the disclosures.

**Table 2.** Results of total and level-1 dimension analysis on social media.

|  | Variable | N | Mean | Ratio | Sd | Min | P50 | Max |
|---|---|---|---|---|---|---|---|---|
| (1) | WeChat_All | 36 | 103.3 | 100.00% | 97.87 | 2 | 72.5 | 470 |
| (2) | ep_w | 36 | 8.14 | 7.88% | 12.14 | 0 | 2 | 50 |
| (3) | cu_w | 36 | 27.06 | 26.20% | 39.61 | 0 | 8.5 | 171 |
| (4) | em_w | 36 | 24.42 | 23.64% | 40.2 | 0 | 5 | 218 |
| (5) | su_w | 36 | 0.64 | 0.62% | 1.33 | 0 | 0 | 6 |
| (6) | co_w | 36 | 5.17 | 5.00% | 6.35 | 0 | 3 | 27 |
| (7) | go_w | 36 | 8.78 | 8.50% | 14.17 | 0 | 3 | 67 |
| (8) | sh_w | 36 | 29.14 | 28.21% | 36.42 | 0 | 18.5 | 141 |

4.1.2. Analysis of the Level-2 Environment Subdimension Analysis on Social Media

Table 3 reports the results of the environment level-2 subdimension analysis on social media. Row (1) of Table 3 shows the statistical results of environment level-1 variable ep_w, which are also reported in Row (2) of Table 2. Rows (2) to (13) show the statistics for environment level-2 subdimension variables, including ep1_w (see Table 1 for details). Among them, the top four level-2 subdimensions with the highest proportions in social media environment disclosure are the "Research, development, application, and sale of the environment production and devices" variable ep8_w, the "Environmental protection investment" variable ep4_w, the "Other environment-related" variable ep0_w, and the "Reduce pollution and decrease drain" variable ep10_w, with mean values of 2.19, 1.25, 1.14, and 0.94, respectively, and the last four are the "Forest certification" variable ep6_w, the "Sustainable forest management" variable ep5_w, the "Quantity, kind, and risk to human and environment of toxic or exhaust emission" variable ep7_w, and the "Environmental impact assessment of new investment projects" variable ep2_w. Column (4) shows

the proportion of level-2 subdimensions to the total disclosure of environment dimensions, e.g., the social media CSR disclosure of the "Environmental management system (EP1)" dimension accounts for 8.23% of the environmental dimension disclosure. Among the twelve subdimensions, the top three ones with the highest percentages are the "Research, development, application, and sale of the environment production and devices dimension (EP8)", the "Environmental protection investment dimension (EP4)", and the "Other environment-related dimension (EP0)", with 26.90%, 15.36%, and 14.00%, respectively. These three types of level-2 subdimensions account for approximately 60% of the total information in the social media environment disclosure.

**Table 3.** Results of environment level-2 subdimension analysis on social media.

| | Variable | N | Mean | Ratio | Sd | Min | P50 | Max |
|---|---|---|---|---|---|---|---|---|
| (1) | ep_w | 36 | 8.14 | 100.00% | 12.14 | 0 | 2 | 50 |
| (2) | ep1_w | 36 | 0.67 | 8.23% | 1.2 | 0 | 0 | 5 |
| (3) | ep2_w | 36 | 0.08 | 0.98% | 0.5 | 0 | 0 | 3 |
| (4) | ep4_w | 36 | 1.25 | 15.36% | 3.15 | 0 | 0 | 12 |
| (5) | ep5_w | 36 | 0.14 | 1.72% | 0.59 | 0 | 0 | 3 |
| (6) | ep6_w | 36 | 0.19 | 2.33% | 0.52 | 0 | 0 | 2 |
| (7) | ep7_w | 36 | 0.11 | 1.35% | 0.4 | 0 | 0 | 2 |
| (8) | ep8_w | 36 | 2.19 | 26.90% | 3.4 | 0 | 1 | 12 |
| (9) | ep9_w | 36 | 0.72 | 8.85% | 2.01 | 0 | 0 | 11 |
| (10) | ep10_w | 36 | 0.94 | 11.55% | 2.29 | 0 | 0 | 10 |
| (11) | ep11_w | 36 | 0.28 | 3.44% | 0.74 | 0 | 0 | 3 |
| (12) | ep12_w | 36 | 0.42 | 5.16% | 1.13 | 0 | 0 | 6 |
| (13) | ep0_w | 36 | 1.14 | 14.00% | 2.18 | 0 | 0 | 10 |

### 4.1.3. Analysis of the Level-2 Customer Subdimension Analysis on Social Media

Table 4 reports the results of the level-2 customer subdimension analysis on social media. Similarly, we report the level-1 customer variable cu_w in Row (1) of Table 4. Rows (2) to (6) of Table 4 show the statistics of level-2 customer variables, including cu1_w. The highest is the "Information provision of the product and services" variable cu4_w, with a mean value of 23.22. Column (4) shows the proportion of level-2 subdimensions to the total disclosure of customer dimensions, e.g., the "Product quality management system (CU1)" subdimension accounts for 2.99% of the whole customer dimension. Among all level-2 customer subdimensions, the highest percentage belongs to the "Information provision of the product and services subdimension (CU4)", whose percentage is 85.81%, accounting for most of the customer dimension disclosures. The above results show that when disclosing customer dimension information on social media, forestry firms prefer to disclose information under the subdimension "Information provision of the product and services (CU4)".

**Table 4.** Results of the level-2 customer subdimension analysis on social media.

| | Variable | N | Mean | Ratio | Sd | Min | P50 | Max |
|---|---|---|---|---|---|---|---|---|
| (1) | cu_w | 36 | 27.06 | 100.00% | 39.61 | 0 | 8.5 | 171 |
| (2) | cu1_w | 36 | 0.81 | 2.99% | 1.69 | 0 | 0 | 8 |
| (3) | cu2_w | 36 | 0.39 | 1.44% | 1.4 | 0 | 0 | 8 |
| (4) | cu4_w | 36 | 23.22 | 85.81% | 37.3 | 0 | 5 | 169 |
| (5) | cu6_w | 36 | 0.33 | 1.22% | 1.2 | 0 | 0 | 6 |
| (6) | cu0_w | 36 | 2.31 | 8.54% | 3.54 | 0 | 0.5 | 13 |

### 4.1.4. Analysis of the Level-2 Employee Subdimension on Social Media

Table 5 reports the results of the analysis of the level-2 employee subdimension on social media. Row (1) of Table 5 shows the statistics of the level-1 variable em_w. Rows (2) to (8) of Table 5 show seven employee level-2 variables. Among them, the top three

variables with the highest scores are "Staff relation management" variable em7_w, "Occupational health and safe producing" variable em6_w, and "Staff development training" variable em5_w, with mean values of 18.25, 3.19, and 2.58, respectively. Column (4) shows the proportion of subdimensions of level-2 disclosure to the total employee dimension disclosure. Among the seven employee subdimensions, the highest disclosure subdimension is the "Staff relation management dimension (EM7)", whose disclosure ratios are 74.73%, accounting for approximately three quarters of the employee dimension disclosure. The above results show that when disclosing employee dimension information on social media, forestry firms prefer to disclose information on the "Staff relation management dimension (EM7)".

**Table 5.** Results of the analysis of the level-2 employee subdimension on social media.

|     | Variable | N | Mean | Ratio | Sd | Min | P50 | Max |
|-----|----------|---|------|-------|-----|-----|-----|-----|
| (1) | em_w | 36 | 24.42 | 100.00% | 40.2 | 0 | 5 | 218 |
| (2) | em1_w | 36 | 0.08 | 0.33% | 0.37 | 0 | 0 | 2 |
| (3) | em3_w | 36 | 0.17 | 0.70% | 0.74 | 0 | 0 | 4 |
| (4) | em4_w | 36 | 0.06 | 0.25% | 0.33 | 0 | 0 | 2 |
| (5) | em5_w | 36 | 2.58 | 10.57% | 6.81 | 0 | 0 | 39 |
| (6) | em6_w | 36 | 3.19 | 13.06% | 6.06 | 0 | 0 | 31 |
| (7) | em7_w | 36 | 18.25 | 74.73% | 28.33 | 0 | 4.5 | 148 |
| (8) | em0_w | 36 | 0.08 | 0.33% | 0.37 | 0 | 0 | 2 |

### 4.1.5. Analysis of the Level-2 Supplier Subdimension on Social Media

Table 6 reports the results of the analysis of the level-2 supplier subdimension on social media. Row 1 shows the statistics of the level-1 variable su_w. Rows (2) to (6) show the results of level-2 supplier variables, in which the top three highest disclosure positions are the "Other supplier-related" variable su0_w, the "Openness of procurement policy" variable su4_w, and the "Legality of forest product procurement" variable su5_w, with mean (median) values of 0.33 (0), 0.14 (0), and 0.11 (0), respectively. Column (4) shows the percentages of the level-2 supplier dimension. The three highest supplier subdimensions are the "Other supplier-related dimension (SU0)", the "Openness of procurement policy dimension (SU4)", and the "Legality of forest product procurement dimension (SU5)", whose disclosure proportions were 51.56%, 21.88%, and 17.19%, respectively, accounting for approximately 90% of the supplier dimension disclosure. The above results indicate that on social media, forestry firms prefer to disclose information on the "Other supplier-related dimension (SU0)", the "Openness of procurement policy dimension (SU4)", and the "Legality of forest product procurement dimension (SU5)", and that these three subdimensions account for approximately 90% of the supplier dimension cumulatively.

**Table 6.** Results of the analysis of the level-2 supplier subdimension on social media.

|     | Variable | N | Mean | Ratio | Sd | Min | P50 | Max |
|-----|----------|---|------|-------|-----|-----|-----|-----|
| (1) | su_w | 36 | 0.64 | 100.00% | 1.33 | 0 | 0 | 6 |
| (2) | su1_w | 36 | 0.03 | 4.69% | 0.17 | 0 | 0 | 1 |
| (3) | su4_w | 36 | 0.14 | 21.88% | 0.42 | 0 | 0 | 2 |
| (4) | su5_w | 36 | 0.11 | 17.19% | 0.4 | 0 | 0 | 2 |
| (5) | su6_w | 36 | 0.03 | 4.69% | 0.17 | 0 | 0 | 1 |
| (6) | su0_w | 36 | 0.33 | 51.56% | 1.01 | 0 | 0 | 5 |

### 4.1.6. Analysis of the Level-2 Community Subdimension on Social Media

Table 7 reports the results of the analysis of the level-2 community subdimension on social media. Row (1) of Table 7 shows the level-1 community dimension variable co_w. Rows (2) to (5) of Table 7 show the statistics for level-2 community variables. Among them, the highest score is the "Effect of enterprise operation on community" variable co1_w,

whose mean value is 3.31. Column (4) shows the proportion of level-2 subdimensions to the level-1 community dimension. The "Effect of enterprise operation on community dimension (CO1)" accounts for 64.02% of the disclosure of community dimensions. The above results indicate that forestry firms are more likely to disclose information on the subdimension "Effect of enterprise operation on community (CO1)" on social media.

**Table 7.** Results the analysis of the level-2 community subdimension on social media.

|  | Variable | N | Mean | Ratio | Sd | Min | P50 | Max |
|---|---|---|---|---|---|---|---|---|
| (1) | co_w | 36 | 5.17 | 100.00% | 6.35 | 0 | 3 | 27 |
| (2) | co1_w | 36 | 3.31 | 64.02% | 4.08 | 0 | 2 | 14 |
| (3) | co4_w | 36 | 0.56 | 10.83% | 1.83 | 0 | 0 | 10 |
| (4) | co6_w | 36 | 0.67 | 12.96% | 1.41 | 0 | 0 | 6 |
| (5) | co0_w | 36 | 0.64 | 12.38% | 1.64 | 0 | 0 | 9 |

### 4.1.7. Analysis of the Level-2 Government Subdimension on Social Media

Table 8 reports the results of the analysis of the level-2 government subdimension. Row (1) of Table 8 shows the statistics of the total government dimension disclosure variable go_w. Rows (2) to (5) of Table 8 show all level-2 subdimensions of government, where the top two highest disclosures are the "Other government-related" variable go0_w and the "Enterprise management abided by rule" variable go1_w, and their mean values are 6.47 and 1.86, respectively. Column (4) shows the proportion of the level-2 subdimensions of the government dimensions. The top two highest disclosure percentages are the "Other government-related dimension (GO0)" and the "Enterprise management abided by rule dimension (GO1)", whose disclosure percentages are 73.69% and 21.18%, respectively, accounting for approximately 95% of the government dimension disclosures.

**Table 8.** Results of the analysis of the level-2 government subdimension on social media.

|  | Variable | N | Mean | Ratio | Sd | Min | P50 | Max |
|---|---|---|---|---|---|---|---|---|
| (1) | go_w | 36 | 8.78 | 100.00% | 14.17 | 0 | 3 | 67 |
| (2) | go1_w | 36 | 1.86 | 21.18% | 3.21 | 0 | 0 | 13 |
| (3) | go2_w | 36 | 0.33 | 3.76% | 0.99 | 0 | 0 | 5 |
| (4) | go4_w | 36 | 0.11 | 1.25% | 0.46 | 0 | 0 | 2 |
| (5) | go0_w | 36 | 6.47 | 73.69% | 11.22 | 0 | 1.5 | 52 |

### 4.1.8. Analysis of the Level-2 Shareholder Subdimension on Social Media

Table 9 reports the results for the shareholder dimension. Row (1) of Table 9 shows the statistical results of the total shareholder dimension variable sh_w. Rows (2) to (6) of Table 9 show the level-2 shareholder variables. Among them, the top three highest disclosures are the "Growth potential" variable sh2_w, the "Other shareholder-related" variable sh0_w, and the "Investor relation management" variable sh1_w, with mean values of 12.89, 7.28, and 4.56, respectively. Column (4) shows the proportion of the level-2 subdimensions. Among the five shareholder subdimensions, the top three highest percentages are the "Growth potential dimension" (SH2), "Other shareholder-related dimension" (SH0), and "Investor relation management dimension" (SH1), whose disclosure percentages are 44.23%, 24.98%, and 15.65%, respectively, accounting for approximately 85% of the shareholder dimension disclosure.

**Table 9.** Results of shareholder level-2 subdimension analysis on social media.

| | Variable | N | Mean | Ratio | Sd | Min | P50 | Max |
|---|---|---|---|---|---|---|---|---|
| (1) | sh_w | 36 | 29.14 | 100.00% | 36.42 | 0 | 18.5 | 141 |
| (2) | sh1_w | 36 | 4.56 | 15.65% | 7.58 | 0 | 2.5 | 42 |
| (3) | sh2_w | 36 | 12.89 | 44.23% | 16.59 | 0 | 7.5 | 68 |
| (4) | sh3_w | 36 | 2.86 | 9.81% | 4.52 | 0 | 1 | 20 |
| (5) | sh4_w | 36 | 1.56 | 5.35% | 3.36 | 0 | 0 | 13 |
| (6) | sh0_w | 36 | 7.28 | 24.98% | 10.06 | 0 | 4.5 | 41 |

4.1.9. Summary of the Analysis of CSR Disclosures on Social Media

First, forestry firms pay significant attention to the interests of different stakeholders on social media. The three most concerned stakeholders on social media CSR disclosure by forestry firms are shareholders (28.21%), customers (26.20%), and employees (23.64%), and these three categories account for approximately 80% of all disclosures.

Moreover, under the level-1 dimension of each stakeholder, forestry firms also show serious disclosure imbalance for different level-2 subdivisions. Specifically, the three level-2 subdimensions with the highest levels of disclosure of the environmental dimension are the "Research, development, application, and sale of the environment production and devices (EP8)" (26.90%), the "Environmental protection investment dimension (EP4)" (15.36%), and the "Other environment-related dimension (EP0)" (14.00%), which account for approximately 60% of the environmental dimension disclosures. The highest reporting subdimensions of the customer and employee dimensions are the "Information provision of the product and services dimension (CU4) (85.81%) and the "Staff relation management" dimension (EM7) (74.73%). The three highest subdimensions in the supplier dimension are the "Other supplier-related dimension (SU0)" (51.56%), the "Openness of procurement policy dimension (SU4)" (21.88%), and the "Legality of forest product procurement dimension (SU5)" (17.19%), accounting for approximately 90% of the supplier dimension disclosures. The highest community subdimension is the "Effect of enterprise operation on community dimension (CO1)" (64.02%). The highest governmental subdimensions are the "Other government-related dimension (GO0)" (73.69%) and the "Enterprise management abided by rule dimension (GO1)" (21.18%), accounting for approximately 95% of the government dimension. The three highest disclosed subdimensions of the shareholder dimension are the "Growth potential dimension (SH2)" (44.23%), the "Other shareholder-related dimension (SH0)" (24.98%), and the "Investor relation management dimension (SH1)" (15.65%), accounting for approximately 85% of the shareholder dimension.

4.2. *Comparative Analysis of CSR Disclosure between Social Media and CSR Reports*

4.2.1. Total and Level-1 Dimension Analysis on CSR Reports

Table 10 reports the descriptive statistics of the total and level-1 dimension variables of the CSR report disclosures for the forestry firms using the WeChat official account (11 of 36 listed forestry firms that use WeChat official account issued the CSR reports in 2018 year). Row (1) of Table 10 shows the statistics of the total disclosure variable Report_All with mean and median values of 26.09 and 23, respectively. Rows (2) to (8) of Table 10 show the statistics of the environment dimension variable ep_r, customer dimension variable cu_r, employee dimension variable em_r, supplier dimension variable su_r, community dimension variable co_r, government dimension variable go_r, and shareholder dimension variable sh_r, where the top three level-1 dimension variables are the environmental variable ep_r, the employee variable em_r, and the supplier variable su_r, with mean (median) values of 6.18 (5), 5.46 (5), and 5.27 (6), respectively. Moreover, their percentages of disclosure are 23.69%, 20.91%, and 20.21%, which in total account for approximately 65% of all disclosures of CSR reports.

**Table 10.** Results of total and level-1 dimension analysis on CSR reports.

| | Variable | N | Mean | Ratio | Sd | Min | P50 | Max |
|---|---|---|---|---|---|---|---|---|
| (1) | Report_All | 11 | 26.09 | 100% | 7.739 | 20 | 23 | 46 |
| (2) | ep_r | 11 | 6.18 | 23.69% | 3.156 | 2 | 5 | 12 |
| (3) | cu_r | 11 | 2.27 | 8.71% | 1.737 | 0 | 2 | 6 |
| (4) | em_r | 11 | 5.46 | 20.91% | 2.339 | 3 | 5 | 10 |
| (5) | su_r | 11 | 1.27 | 4.88% | 1.104 | 0 | 1 | 3 |
| (6) | co_r | 11 | 2.91 | 11.15% | 1.3 | 1 | 4 | 4 |
| (7) | go_r | 11 | 2.73 | 10.45% | 2.37 | 0 | 2 | 6 |
| (8) | sh_r | 11 | 5.27 | 20.21% | 2.453 | 2 | 6 | 8 |

### 4.2.2. Comparison between CSR Disclosure under Social Media and CSR Reports

In Figure 1, we compare forestry firms' disclosures on social media and CSR reports by stakeholder dimensions.

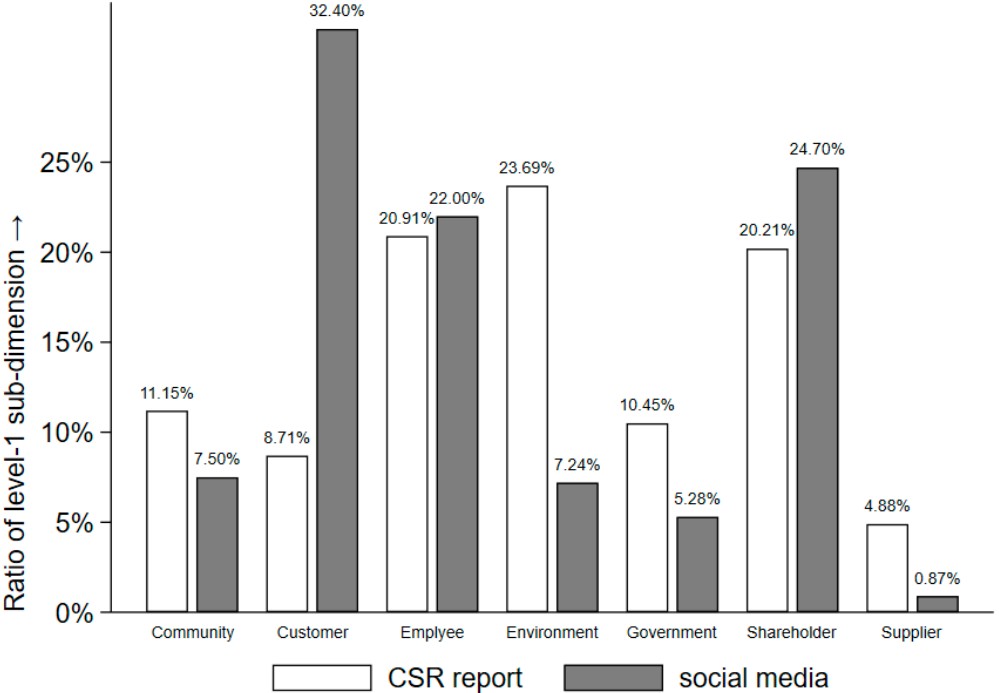

**Figure 1.** Comparison of CSR disclosure on social media and CSR reports.

First, there are differences in the stakeholder dimensions that forestry firms focus on in different channels. The top three dimensions in CSR report channels are the environment (23.69%), employee (20.91%), and shareholder (20.21%) dimensions, while the top three dimensions in social media channels are the customer (32.40%), shareholder (24.70%), and employee (22.00%) dimensions. The dimension with the highest proportion disclosed in the report is the environment dimension (23.69%), but this dimension only ranks 5th on social media (7.24%), with a difference of more than three times, while the customer dimension, which ranks 6th in the CSR report in terms of proportion (8.71%) and 1st on social media (32.40%), has the same difference of more than three times. We suggest that this difference may be due to the characteristics of the information users focused on by firms in different channels: forestry firms are mostly in heavily polluting and environmentally sensitive industries, so in formal reports (CSR reports), they take more into account the needs of professional information users, such as providing more detailed and rich environmental information to analysts, socially responsible investors (SRIs), fund managers, etc. However, in social media, forestry firms are more likely to have to communicate with non-specialist users, e.g., local customers.

Second, there are also commonalities in the stakeholder dimensions that forestry firms focus on under different channels. For example, in both reporting channels, the lowest proportion of disclosure is the supplier dimension (with CSR reports at 4.88% and social media at 0.87%). This is because forestry firms mostly belong to the upstream of the industry chain or self-supply their raw materials, such as paper, wood supply, furniture manufacturing, and other niche industry firms, and suppliers have little influence on them. In addition, the proportion of community dimension and government dimension is also small, and the proportion in the CSR reports and social media is mostly around 10% or lower. This indicates that for Chinese forestry firms, these two types of stakeholders are not the focus of corporate concern.

## 5. Discussion and Conclusions

This paper studies the CSR disclosures made by Chinese forestry firms on social media. Based on the framework of Lu, Kozak, Toppinen, D Amato, and Wen [41], we used content analysis to analyze the social media posted by Chinese-listed forestry firms on WeChat in 2018 and found that (1) there are differences for Chinese forestry firms that care about stakeholders in social media and traditional CSR report channels; for example, the top three dimensions present in the social media channel are shareholder, customer, and employee dimensions, while the top three dimensions present in the traditional CSR report channel are environment, employee, and shareholder dimensions; (2) the proportion of disclosed subcontent for each stakeholder on social media also shows a great imbalance; for example, the most concerning subdimension in the disclosure for the shareholder dimension is the "Growth potential" subdimension, which accounted for 44.23% of the shareholder's information. The results of our analysis suggest that the characteristics of social media users lead to distinctive features of CSR disclosures made by forestry firms on social media, with significant differences in the main points of attention from disclosures made through traditional channels (CSR reports). We argue that the main reasons for the differences in the focus of information disclosure between forestry firms in social media and traditional CSR reports are as follows: the target information users of CSR reports are professional users, e.g., analysts, institutional investors, and financial media; however, the general public (e.g., individual consumers, grassroots employees, and community residents) constitute the largest proportion of stakeholders but are less capable of using information, specifically, they lack intuitive judgment of the financial indicators [43] and industrial technical indicators recorded in traditional reports, and are more concerned with intuitive information such as protection of consumer rights and demonstration of corporate strengths, so forestry firms are more interested in displaying such intuitive information in social media than in CSR reports.

The following questions of this paper need future research: first, the WeChat platform cannot fully display all the comments because of its program design (According to the rules of the WeChat platform, users' comments on WOA need to be reviewed by the account owner before they are made public. As a result, negative comments about the firm are filtered out), which leads us to be unable to analyze the interactive communication between firms and information users; second, information activities conducted by firms on the WeChat platform use a large number of pictures and videos, and Kassinis and Panayiotou [44] found that visualization tools play an important role in CSR impression management by firms, which is limited by technical means, making it difficult for us to include these images and videos in our analysis; third, some non-listed forestry firms also use social media to communicate CSR messages, but we are unable to include them in our analysis due to the difficulty of obtaining their firm-level data. We look forward to obtaining additional information through questionnaire analyses, field research, and case studies in the future to provide further answers to the above questions. Furthermore, we will investigate the following in future studies: (1) the motivation and the factors affected on the forestry firms' disclosure differences between social media and CSR report; (2) So-

cial media CSR disclosure of other industries related to carbon emission reduction (e.g., energy industry [45]).

**Author Contributions:** M.Z.: conceptualization, design, manuscript preparation. F.L.: conceptualization, design, manuscript preparation and supervised this project. Y.Z. and J.C.: data preparation and design, formal analysis, original draft preparation and manuscript preparation sections. Each author contributed to the conceptualization and writing of this paper. All authors have read and agreed to the published version of the manuscript.

**Funding:** This research was funded by the National Natural Science Foundation of China (Grant number 71902090), Forestry Science and Technology Development Project of China (Grant number KJZXRZ202013), the University Innovation and Technology Project of Shandong Province in China (Grant number 2020RWG004), the High-level Research Foundation of QAU (Grant number 1119710), and the Project Supported by Enterprises and Institutions (Grant number 6602422730).

**Institutional Review Board Statement:** This study does not involve any ethical issues.

**Data Availability Statement:** The data of this research are publicly available.

**Conflicts of Interest:** The authors declare no conflict of interest.

## Appendix A

**Table A1.** CSR content analysis system and examples of social media from WeChat Official Account.

| Level-1 Indicators. | Definitions | Level-2 Indicators | Definitions | Keywords for Identification | Examples of Scoring |
|---|---|---|---|---|---|
| | | EP1 | Environment management system | Environmental management objectives, environment protection certification for firms, low-carbon green certification and awards, building eco-industrial chains, et al. | Chenming Paper (000488) reported: "Chenming Group was the first in its industry to pass the ISO14001 environmental management system certification in China". We marked 1 point for the EP1. |
| | | EP2 | Environmental impact assessment of new investment project | New construction projects are expected to reduce pollution emissions and are expected to improve the environment of a region, et al. | Yuntou Ecology (002200) reported: "After the project is completed, it can effectively collect and treat domestic sewage from the new urban area of Tonghai County, Xiushan Street, Jiulong Street, Sijie Town, and other areas, reducing the pollution caused by domestic sewage discharged directly into Qilu Lake". We marked 1 point for the EP2. |
| | | EP3 | Forest biodiversity conservation | n. a. | n. a. |
| | | EP4 | Environment protection investment | Amount of investment in environmental protection projects, amount of investment in environmental protection projects, et al. | Chenming Paper (000488) reported: "We have invested more than 8 billion RMB in environmental protection projects, taking the green low-carbon cycle development path". We marked 2 points for the EP4. |
| | | EP5 | Sustainable forest management | Conservation of forest resources, et al. | Yueyang Forest and Paper (600963) reported: "We placed the 'first workshop' in the forested hills to feed the forest with paper and promote paper with the forest to achieve the effect of flourishing forest and paper". We marked 1 point for the EP5. |
| | | EP6 | Forest certification | Forest stewardship council, et al. | Sun Paper (002078) reported: "FSC-certified 'Happy Sunshine' household paper received positive feedback from the public". We marked 1 point for the EP6. |

**Table A1.** *Cont.*

| Level-1 Indicators. | Definitions | Level-2 Indicators | Definitions | Keywords for Identification | Examples of Scoring |
|---|---|---|---|---|---|
| EP | Environment Protection | EP7 | The quantity, kind, and risk to human and environment of toxic or exhaust emission | Type or number of harmful substances emitted, et al. | Fenglin Group (601996) reported: "Exhaust gas and wastewater emissions are better than national standards, especially the particulate matter emission concentration is about 3.5mg/mL". We marked 2 points for the EP7. |
| | | EP8 | Research, development, application, and sale of the environment production and devices | Opening and production of environmentally friendly products, application of environmental technology, renewal of environmental protection equipment, environmental certification of products, production of environmental protection, recycling of old furniture and other forest products, et al. | Chenming Paper (000488) reported: "The company is actively promoting the application of zero-water discharge technology and has visited several countries to learn about wastewater treatment and recycling technology". We marked 1 point for the EP8. |
| | | EP9 | Energy resources conservation | Save heavy oil, save water, save electricity, save paper resources, et al. | Chenming Paper (000488) reported: "We are the first one in the industry to put into operation a water reuse project, with a reuse rate of over 40% and water consumption per ton of paper reduced to less than half of the international standard level". We marked 2 points for the EP9. |
| | | EP10 | Reduce pollution and decrease drain | Reduction of carbon emissions, reduction of emissions of other gases and substances, reuse of waste, et al. | Yueyang Forest and Paper (600963) reported: "We made construction of deep sewage treatment upgrade project and fluidized bed boiler ultra-low emission project, with a significant reduction in sewage COD and nitrogen oxide emission concentration". We marked 1 point for the EP10. |
| | | EP11 | Ecology restoration | Forest restoration, land loss reduction, ecological protection, et al. | Chenming Paper (000488) reported: "Retirement of the forest will have created 3.25 million mu of completed forest land all returned to Huanggang, not cutting a tree in the old areas, production of raw materials is imported wood chips". We marked 2 points for the EP11. |
| | | EP12 | Volunteer working for environmental protection | External sanitation and cleaning work, tree planting activities, et al. | Fujian Jinsen (002679) reported: "When the 33rd International Volunteer Day, the Party Committee of Jinsen Group organized volunteers to carry out "clean up the home" environmental protection volunteer work in the co-construction community—Dongmen Community ". We marked 1 point for the EP12. |
| | | EP0 | Other environment-related | Green and sustainable development strategies, promotion of green and environmental protection concepts, environmental education, elimination of backward production capacity, et al. | Chenming Paper (000488) reported: "Chenming Group has eliminated 2.72 million tons of backward production capacity, with an elimination rate of 27%". 2 points for the EP0. |

**Table A1.** *Cont.*

| Level-1 Indicators. | Definitions | Level-2 Indicators | Definitions | Keywords for Identification | Examples of Scoring |
|---|---|---|---|---|---|
| CU | Customers | CU1 | Product quality management system | Quality supervision, measurement management system in production department, standardized operation, quality management system certification, FSC-COC chain-of-custody system certification, et al. | Sophia (002572) reported: "Sophia has got ISO 9001 quality management system certification multiple times". We marked 1 point for the CU1. |
| | | CU2 | After-sale service system | Content of after-sales service, customer satisfaction, et al. | Sophia (002572) reported: "From the preparation of materials, the plate has been attached to the exclusive QR code, and this QR code will be the identity card of this piece of plate, and after that, sealing, punching, packaging, and other links, until the terminal sales, all the information of the original plate can be tracked at any time. This is more conducive to ensuring stable quality and good after-sales service". We marked 1 point for the CU2. |
| | | CU3 | Dispute settlement mechanism | n. a. | n. a. |
| | | CU4 | Information provision of the product and services | Product exhibitions, marketing information, display and promotion of products and services and design concepts, product advertising information, et al. | Qumei Home Furnishings (603818) reported: "During the 3rd season of the national sofas sales promotion, Qumei's new fashion series of sofa start from as low as 3999 RMB, with a variety of colors and moods for customers". We marked 2 points for the CU4. |
| | | CU5 | Privacy protection of customer | n. a. | n. a. |
| | | CU6 | Forest products and other green marketing | Green communication, green marketing activities for forest products, et al. | Qumei Home Furnishings (603818) reported: "The national launch ceremony of Qumei Home furnishings' 'Trade-in' sixth season large green series activities were successfully held at the '2018 Beijing International Home Furnishings Exhibition and Intelligent Life Festival'. Immediately afterward, the green life advocated by 'Trade-in' swept the country with a prairie momentum, and dealers in over 300 cities responded positively". We marked 2 points for the CU6. |
| | | CU0 | Other customer-related | Customer relationship management activities, dealer conferences, customer visits and communications, customer evaluations of products or services, and other information on customer-related activities, et al. | Fujian Jinsen (002679) reported: "On the morning of August 3, in the 9th-floor conference room of Hua Hong Technology, the company's fourth logging area timber production and sales tender will be successfully concluded. All the publicized bids were invited, with an area of 1522 m$^2$ and a timber volume of 12,103 m$^3$, and the bid amounted to 9.5 million RMB, an increase of 21.3% over the previous year". We marked 2 points for the CU0. |

**Table A1.** *Cont.*

| Level-1 Indicators. | Definitions | Level-2 Indicators | Definitions | Keywords for Identification | Examples of Scoring |
|---|---|---|---|---|---|
| EM | Employees | EM1 | Abidance by rule and laws | Enterprise compliance with labor laws, et al. | Minfeng Special Paper (600235) reported: "The third staff congress of the company was held in the East Conference Room of the Administration Building, and more than 110 staff representatives from all units and departments of the company attended the meeting. The General Assembly considered a new round of Collective Contract. Since the last round of Collective Contract was signed, the Company has been able to strictly implement the content of the terms and conditions of the contract, and no violation of the contract has occurred in the past three years". We marked 2 points for the EM1. |
| | | EM2 | Percent of contract signing | n. a. | n. a. |
| | | EM3 | Coverage of social insurance | Employee social insurance, et al. | Yutong Technology (002831) reported: "Does the Yuxinyutong Personnel Workers Commercial Insurance take effect immediately after taking out the policy? Sickness death and total disability and critical illness are subject to a 30-day waiting period, and cases occurring within 30 days of enrollment will not be paid. Accidental death and disablement are effective immediately". We marked 1 point for the EM3. |
| | | EM4 | Equal employment institution | Male and female employee ratio, et al. | Chenming Paper (000488) reported, "Chenming has more than 3160 female employees, accounting for 31% of the total staff". We marked 2 points for the EM4. |
| | | EM5 | Staff development training | Staff training (skill-based), staff training seminars and events, et al. | Yutong Technology (002831) reported: "The Group's Human Resources Management Center brought the course to Vietnam on August 24–August 25. Thirty-six grassroots management cadres (managers, section chiefs and reserve section chiefs) from Yutong Vietnam and Yuzhan Vietnam attended the training, with an attendance rate of 90%". We marked 2 points for the EM5. |
| | | EM6 | Occupational health and safe producing | Improvement of work and production environment, employee medical check-ups, safety education and drills, safety hazard inspections, operational and production safety regulations, et al. | Mcc Meili Paper (000815) reported: "From April 2-4, Meili Cloud organized relevant leaders and safety managers, totaling 70 people, to conduct safety qualification certificate review and certification training at the Zhongwei Safety Production Training Center". We marked 2 points for the EM6. |
| | | EM7 | Staff relation management | Commendation of employees, recruitment, senior management retirement, employee cultural and entertainment activities, employee care, condolence to employees, corporate culture construction, employee stock ownership, firm internal journals, et al. | Yuntou Ecology (002200) reported: "This staff sports game had 31 participating delegations, more than 860 athletes and 60 referees, as well as more than 100 staff performance teams and 60 volunteer teams, which was a high participation, high quality, and high-level sports event". We marked 2 points for the EM7. |

**Table A1.** *Cont.*

| Level-1 Indicators | Definitions | Level-2 Indicators | Definitions | Keywords for Identification | Examples of Scoring |
|---|---|---|---|---|---|
| | | EM0 | Other employee-related | Promotion of academic qualifications, et al. | Fenglin Group (002078) reported: "Eleven employees of Fenglin factory were happy to get their Adult Education undergraduate certificates, and everyone happily took a group photo in front of the company". We marked 2 points for the EM0. |
| SU | Suppliers | SU1 | Responsibility purchasing system | Responsible purchasing, et al. | Minfeng Special Paper (600235) reported: "The company sent an auditor to conduct a comprehensive audit of the purchasing, production, inventory and sales practices of FSC products produced by Minfeng Special Paper in the past year". We marked 1 point for the SU1. |
| | | SU2 | Credit rating | n. a. | n. a. |
| | | SU3 | Contradict performance rate | n. a. | n. a. |
| | | SU4 | The openness of procurement policy | Number and amount of material procurement, procurement project signing ceremony, et al. | Chenming Paper (000488) reported: "The raw materials for the project are purchased and self-made cellulose-dissolving wood pulp. The purchased portion is procured uniformly in the market through the procurement channel of the group company; the self-made pulp is provided by the existing pulp-making project under construction, which has stable production and can ensure a reliable supply of fiber raw materials for the project". We marked 1 point for the SU4. |
| | | SU5 | The legality of forest product procurement | Wood procurement information, paper procurement information, et al. | Mcc Meili Paper (000815) reported, "We expect to import about 15.5 million tons of waste paper for the year, down about 10 million tons from last year". We marked 2 points for the SU5. |
| | | SU6 | Supplier qualification evaluation | Selection of high-quality suppliers, et al. | Sophia (002572) reported: "Mr. Jiang Ganjun, Chairman of the Board of Directors, Mr. Ke Jiansheng, President of the Company, and other senior executives presented awards to outstanding suppliers on behalf of the Company". We marked 1 point for the SU6. |
| | | SU0 | Other supplier-related | Bidding information, supplier conferences, supplier relationship management, et al. | Sophia (002572) reported: "In 2018 Annual Sophia Home Supplier Conference, nearly 600 supplier representatives attended the conference to discuss how to cooperate closely with the upstream and downstream of the home furnishing industry in the context of the new era for win-win development". We marked 2 points for the SU0. |
| | | CO1 | The effect of enterprise operation on community | Promotion of employment, poverty alleviation, disaster relief, product support for local, social public welfare activities, et al. | Chenming Paper (000488) reported, "Zhanjiang Chenming built the largest plant of new wall material in Zhanjiang City, all of which digests solid waste such as ash slag generated from power plants and uses it as raw material to produce lightweight bricks for sale to urban areas, supporting Zhanjiang's urban construction". We marked 1 point for the CO1. |

**Table A1.** *Cont.*

| Level-1 Indicators. | Definitions | Level-2 Indicators | Definitions | Keywords for Identification | Examples of Scoring |
|---|---|---|---|---|---|
| CO | Community | CO2 | Staff localization policy | n. a. | n. a. |
| | | CO3 | Localization procurement policy | n. a. | n. a. |
| | | CO4 | Donations institution and amount | Donations institution and amount, et al. | Jingxing Paper (002067) reported: "Company Chairman Zhu Zailong donated RMB 600,000 to Zhang Lou Primary School on behalf of the company". We marked 2 points for the CO4. |
| | | CO5 | The policy of support for volunteer activity | n. a. | n. a. |
| | | CO6 | The data of staff volunteer activity | Employee volunteer activity records, employee donations, and other employee volunteer activities, et al. | Sun Paper (002078) reported: "Chen Wenjun, the company's vice president, took the lead in making donations for the disaster area, and more than 400 Sun people donated in order". We marked 2 points for the CO6. |
| | | CO0 | Other community-related | Communication and cooperation with local educational institutions, cooperation with local NGOs, et al. | Sun Paper (002078) reported: "A group of 24 people from the School of Light Industry Science and Engineering of Shaanxi University of Science and Technology came to Shandong Sun Paper Co. for exchange and study". We marked 2 points for the CO0. |
| GO | Government | GO1 | Enterprise management abided by rule | Policy support, response to policy calls, clean and anti-corruption, protection of intellectual property rights, et al. | Yueyang Forest and Paper (600963) reported, "Discipline Inspection Committee of Yueyang Forest and Pape issued the Notice on the Implementation Plan of Yueyang Forest and Paper's 2018 'Anti-Corruption and Integrity Propaganda and Education Month' Activities ". We marked 1 point for the GO1. |
| | | GO2 | Tax payment | Tax recognition, tax payment amount, et al. | Yinhua Lifestyle Technology (600978) reported: "Taxes paid are 2.1 billion RMB". We marked 2 points for the GO2. |
| | | GO3 | Employment security policy | n. a. | n. a. |
| | | GO4 | Employment amount over the report periods | Number of employees in the enterprise, et al. | UE Furniture (603600) reports, "The company employs nearly 4500 people". We marked 2 points for the GO4. |
| | | GO0 | Other government-related | Participation in local or national People's Congress, communication and cooperation with the government, governmental officer visits, party organization building, party member activities, et al. | Yuntou Ecology (002200) reported, "On June 30, the Party Committee of Yuntou Ecology organized party education for all party members and cadres of the company to further strengthen their ideal beliefs and awareness of purpose and to remember the responsibilities and obligations of party members by watching the red movie 'Unforgettable Years' and revisiting the historical stories of party building during the Yan'an period". We marked 1 point for the GO0. |

**Table A1.** *Cont.*

| Level-1 Indicators. | Definitions | Level-2 Indicators | Definitions | Keywords for Identification | Examples of Scoring |
|---|---|---|---|---|---|
| SH | Shareholders | SH1 | Investor relation management | Holding of senior management meetings, executive appointments, general meeting of stockholders, corporate investment activities, mergers and acquisitions or creation of subsidiaries, formulation of corporate strategies, strategic cooperation between other enterprises, changes in corporate shares, changes on shareholder holdings, annual planning, financing activities such as issuance of bond, the shareholding of distributors, dividends, et al. | Chenming Paper (000488) reported: "Chenming will acquire 1.369 billion shares of Nanyue Bank with 2.546 billion RMB, accounting for 14.55% of the total share capital of Nanyue Bank, through a series of combined operations such as 'Subscription to privately issued shares + Public purchase' by its subsidiary". We marked 2 points for the SH1. |
| | | SH2 | Growth potential | Project information, production capacity, sales growth, new production lines, technological breakthroughs, description of company status, company development history, talent reserves, et al. | Chenming Paper (000488) reported: "Zhanjiang Chenming invested a total of 26.5 billion yuan to build these four integrated pulp and paper production lines, of which the first phase production line and the fourth phase production line are the cultural paper production lines and white cardboard production lines with the widest paper width, the fastest speed, and the highest single-machine capacity in the world". We marked 2 points for the SH2. |
| | | SH3 | Profitability | Financial indicators such as profit and operating income, brand value, et al. | Chenming Paper (000488) reported: "The project will achieve annual sales revenue of approximately RMB 9904 million and net profit of approximately RMB 1016 million upon completion". We marked 2 points for the SH3. |
| | | SH4 | Safety | Safety of financing, safety of operation, Safety of production materials such as inventory, et al. | Guangdong Ganhua (000576) reported: "The diversified business model would also strengthen the company's financial soundness, enhance the company's anti-risk capability, and help protect the interests of the shareholders, especially the small and medium-sized shareholders". We marked 1 point for the SH4. |
| | | SH0 | Other shareholder-related | Hold or participate in academic activities, external publicity of corporate culture, social interviews of executives, et al. | Yueyang Forest and Paper (600963) reported: "Yueyang Forest and Paper, upholding the essence of history and culture and shouldering the burden of sustainable development, gathered with nearly 400 partners to talk about the future". We marked 2 points for the SH0. |

## Appendix B

**Table A2.** Firms sampled in this study.

| ID | Firms | WeChat Name | Websites of the CSR Report |
|---|---|---|---|
| 000488 | Chenming Paper Group Co., Ltd. | 晨鸣集团 | |
| 000576 | Guangdong Ganhua Co., Ltd. | 广东甘化000576 | |
| 000815 | Mcc Meili Cloud Computing Industry Investment Co., Ltd. | 中冶美利云产业投资股份有限公司 | http://www.cninfo.com.cn/new/disclosure/detail?stockCode=000815&announcementId=1206091798&orgId=gssz0000815&announcementTime, accessed on 25 April 2019. |
| 000833 | Guangxi Guitang (Group) Co., Ltd. | 粤桂股份 | |
| 002067 | Zhejiang Jingxing Paper Co., Ltd. | 景兴纸业002067 | http://www.cninfo.com.cn/new/disclosure/detail?stockCode=002067&announcementId=1206113242&orgId=9900000601&announcementTime, accessed on 27 April 2019. |
| 002078 | Shandong Sun Paper Co., Ltd. | 山东太阳纸业股份有限公司 | http://www.cninfo.com.cn/new/disclosure/detail?stockCode=002078&announcementId=1206020135&orgId=9900001223&announcementTime, accessed on 16 April 2019. |
| 002200 | Yuntou Ecology Co., Ltd. | 云投生态 | |
| 002228 | Yueyang Forest and Paper Co., Ltd. | 合兴包装ips | |
| 002235 | Anne Co., Ltd. | 安妮股份 | |
| 002303 | Meiyingsen Group Co., Ltd. | 美盈森集团 | |
| 002521 | Qifeng New Material Co., Ltd. | 齐峰新材 | |
| 002572 | Sophia Household Co., Ltd. | 索菲亚家居 | http://www.cninfo.com.cn/new/disclosure/detail?stockCode=002572&announcementId=1205874312&orgId=9900019037&announcementTime, accessed on 5 March 2019. |
| 002679 | Fujian Jinsen Forestry Co., Ltd. | 福建金森集团 | |
| 002798 | D&O Home Collection Group Co., Ltd. | 帝王洁具monarch | |
| 002831 | Shenzhen YUTO Packaging Technology Co., Ltd. | 裕同科技 | |
| 002853 | Guangdong Piano Home Co., Ltd. | 皮阿诺家居 | |
| 600076 | Kangxin New Materials Co. Ltd. | 康欣新材料股份有限公司 | |
| 600235 | Minfeng Special Paper Co., Ltd. | 民丰特纸 | |
| 600321 | Rightway Holding Co., Ltd. | 正源股份 | |
| 600356 | Mudanjiang Hengfeng Paper Co., Ltd. | 牡丹江恒丰纸业股份有限公司 | http://www.cninfo.com.cn/new/disclosure/detail?stockCode=600356&announcementId=1206117239&orgId=gssh0600356&announcementTime, accessed on 27 April 2019. |
| 600433 | Guangdong Guanhao High-Tech Co., Ltd. | 广东冠豪高新技术股份有限公司 | http://www.cninfo.com.cn/new/disclosure/detail?stockCode=600433&announcementId=1205903842&orgId=gssh0600433&announcementTime, accessed on 18 March 2019. |
| 600963 | Yueyang Forest and Paper Co., Ltd. | 百年岳纸千载文章 | |
| 600978 | Yinhua Lifestyle Technology Co., Ltd. | 宜华生活创享优悦 | |
| 601996 | Guangxi Fenglin Wood Industry Co., Ltd. | 丰林集团 | http://www.fenglingroup.com/shzrbg/info_31.aspx?itemid=3137, accessed on 18 March 2019. |
| 603022 | Shanghai XTL Packaging Co., Ltd. | 新通联xtl | |
| 603165 | Rongsheng Environmental Protection Technology Co., Ltd. | 荣晟环保 | http://www.cninfo.com.cn/new/disclosure/detail?stockCode=603165&announcementId=1205931681&orgId=9900030004&announcementTime, accessed on 23 March 2019. |

**Table A2.** *Cont.*

| ID | Firms | WeChat Name | Websites of the CSR Report |
|---|---|---|---|
| 603180 | Gold kitchen cabinet Home Technology Co., Ltd. | 金牌厨柜官方号 | |
| 603208 | Oupai Group Co., Ltd. | 欧派 | |
| 603226 | Vohringer Home Technology Co., Ltd. | 菲林格尔vohringer | CSR report provided by China Forest Products Industry Association |
| 603313 | Mlily Furniture Co., Ltd. | mlily 梦百合 | |
| 603326 | OLO Furniture Co., Ltd. | olo 我乐家居 | |
| 603389 | A-Zenith Furniture Co., Ltd. | a-zenith 亚振 | |
| 603600 | UE Furniture Co., Ltd. | 永艺股份 | |
| 603801 | ZBOM Furniture Co., Ltd. | 志邦家居 | |
| 603816 | KUKA Home Co., Ltd. | 顾家家居 | http://www.cninfo.com.cn/new/disclosure/detail?stockCode=603816&announcementId=1206054348&orgId=9900027317&announcementTime, accessed on 19 April 2019. |
| 603818 | Qumei Home Furnishings Group Co., Ltd. | 曲美家居 | http://www.qumei.com/upload/files/2020/3/b259ecb270526e84.pdf, accessed on 19 April 2019. |

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
