# Peer review of "What Corporate Social Responsibility (CSR) Disclosures Do Chinese Forestry Firms Make on Social Media? Evidence from WeChat"

_forests, doi:10.3390/f13111842_

Round 1

Reviewer 1 Report

This paper examines and compares content from Chinese forest companies’ reports and CSR “tweets” ib WeChat and uses content analysis to compare the focus of the messaging.  The sample and methodology seem sound and the results are interesting in that the two styles of communication (report versus social media output) target different audiences and present different messages.

The authors do a good job of reporting the statistics and differences, but I think this paper would be more valuable and insightful if the authors could expand on the implications of the results.  Why might they send different messages? What are forestry firms trying to convey or hide?  I would expect that shareholders care about returns and therefore, messaging would be targeted towards financial outcomes. Similarly, I would expect that the use of social media would with environmental messaging would help to garner support from the public, which might be more concerned about the effects of logging.

Finally, could you comment on why this information is important and/or how it could be used to further investigate the forest industry’s actions?  How does your work contribute to the literature?

Overall, the paper is very well written, easy to follow and methodologically sound.  My recommendation is to extend the implications.

Author Response

A Report of Amendments and Responses to the Editor’s and Reviewers’ Comments

Hope you are fine. We are grateful to you for allowing us to revise and resubmit the paper. We would also like to thank the anonymous reviewer for his/her insightful and constructive comments. We have fully revised the paper in line with all your recommendations, which have helped us in further improving the quality of the current version of the paper. We have outlined below the specific amendments that we have carried out in response to each of your suggestions.

We are grateful for your careful reading of our paper and more importantly, your positive and constructive comments. We have incorporated all your comments and suggestions into this version of the manuscript. Because of your thoughtful review of the manuscript, we believe that our revision has greatly improved the paper. We hope that we have succeeded in addressing the concerns raised in your comments.

Below, we detail our point-by-point responses to your comments (in bold and italic fonts). Our response starts with ***. All revisions are highlighted in the manuscript.

Reviewer 2 Report

Dear authors,

After reading your paper, i find it very interesting and well presented. however, i think that taking into account the following remarks may further enhance the quality of your paper.

- first, in the key words, i suggest to add the term Corporate social responsibility.

- in the introduction, the authors wrote "Related studies only 61 focus on forestry firms' CSR disclosures based on CSR reports but lack consideration of 62 such disclosures on social media". please add references to justify this sentence. besides. you need to justify the chinese the context. why did you choose to study the chinese companies.

- in the literature review, the authors need to add and review some empirical studies. here it is suggested to add the following papers:

Ghardallou W., N. Alessa (2022). Corporate Social Responsibility and Firm Performance in GCC Countries: A Panel Smooth Transition Regression Model. Sustainability, 14(13): 7908

 Ghardallou, W. (2022). Corporate Sustainability and Firm Performance: The Moderating Role of CEO Education and Tenure. Sustainability, 14(6):3513.

- The methodology is clear and well presented.

- Results are well discussed.

- In the conclusion, the authors should add policy and recommendations of the paper.

Author Response

A Report of Amendments and Responses to the Editor’s and Reviewers’ Comments

Hope you are fine. We are grateful to you for allowing us to revise and resubmit the paper. We would also like to thank the anonymous reviewer for his/her insightful and constructive comments. We have fully revised the paper in line with all your recommendations, which have helped us in further improving the quality of the current version of the paper. We have outlined below the specific amendments that we have carried out in response to each of your suggestions.

We are grateful for your careful reading of our paper and more importantly, your positive and constructive comments. We have incorporated all your comments and suggestions into this version of the manuscript. Because of your thoughtful review of the manuscript, we believe that our revision has greatly improved the paper. We hope that we have succeeded in addressing the concerns raised in your comments.

Below, we detail our point-by-point responses to your comments (in bold and italic fonts). Our response starts with ***. All revisions are highlighted in the manuscript.

Reviewer-2 Comments:

Comments to the Author

After reading your paper, I find it very interesting and well presented. The methodology is clear and well presented. Results are well discussed. The following are a summary of my concerns:

1.In the key words, I suggest to add the term Corporate social responsibility.

*** We are grateful for this constructive comment. We have added it in the Keyword.

2.In the introduction, the authors wrote "Related studies only 61 focus on forestry firms' CSR disclosures based on CSR reports but lack consideration of 62 such disclosures on social media". please add references to justify this sentence. besides. you need to justify the Chinese the context. why did you choose to study the Chinese firms.

*** We are grateful for this constructive comment. We have added a note in 3.1 Data sources (Line 161): “As the largest developing economy, China has a huge forestry industry. According to a report by the China National Forestry and Grassland Administration, gross output value of forestry reached 6.49 trillion RMB in 2016 and is expected to reach 9 trillion RMB by 2025. Thus, Chinese forestry firms have a strong representation.”

3.In the literature review, the authors need to add and review some empirical studies. here it is suggested to add the following papers:

Ghardallou W., N. Alessa (2022). Corporate Social Responsibility and Firm Performance in GCC Countries: A Panel Smooth Transition Regression Model. Sustainability, 14(13): 7908

Ghardallou, W. (2022). Corporate Sustainability and Firm Performance: The Moderating Role of CEO Education and Tenure. Sustainability, 14(6):3513.

*** We are grateful for this constructive comment. We have further combed the literature in our paper.

Thank you again for your careful reading of the paper and for pointing out so many important issues. We have carefully revised our paper following your great comments and the new version has improved a lot.

Thank you again for offering the opportunity for revision.

Yours sincerely

Feifei Lu
